# Mesenchymal Stem Cell-Derived Extracellular Vesicles as Proposed Therapy in a Rat Model of Cerebral Small Vessel Disease

**DOI:** 10.3390/ijms231911211

**Published:** 2022-09-23

**Authors:** Reut Guy, Shay Herman, Hadar Benyamini, Tali Ben-Zur, Hila Kobo, Metsada Pasmanik-Chor, Dafna Yaacobi, Eric Barel, Chana Yagil, Yoram Yagil, Daniel Offen

**Affiliations:** 1Department of Human Genetics and Biochemistry, Sackler School of Medicine, Felsenstein Medical Research Center, Tel Aviv University, Tel Aviv 69978, Israel; 2Info-CORE, Bioinformatics Unit of the I-CORE at the Hebrew University, Jerusalem 9103401, Israel; 3Genomics Research Unit, George Wise Faculty of Life Sciences, Tel Aviv University, Tel Aviv 69978, Israel; 4Bioinformatics Unit, George Wise Faculty of Life Sciences, Tel Aviv University, Tel Aviv 69978, Israel; 5Department of Plastic and Reconstructive Surgery, Rabin Medical Center, Petah-Tikva 49100, Israel; 6Israeli Rat Genome Center, Laboratory for Molecular Medicine, Barzilai University Medical Center, Ashkelon 78306, Israel; 7Faculty of Health Sciences, Ben-Gurion University of the Negev, Beer-Sheva 8410501, Israel; 8Sagol School of Neuroscience, Tel Aviv University, Tel Aviv 69978, Israel

**Keywords:** mesenchymal stem cells, extracellular vesicles, cerebral small vessel disease

## Abstract

Mesenchymal stem cell-derived extracellular vesicles (MSC-EVs) have been employed in the past decade as therapeutic agents in various diseases, including central nervous system (CNS) disorders. We currently aimed to use MSC-EVs as potential treatment for cerebral small vessel disease (CSVD), a complex disorder with a variety of manifestations. MSC-EVs were intranasally administrated to salt-sensitive hypertension prone SBH/y rats that were DOCA-salt loaded (SBH/y-DS), which we have previously shown is a model of CSVD. MSC-EVs accumulated within brain lesion sites of SBH/y-DS. An in vitro model of an inflammatory environment in the brain demonstrated anti-inflammatory properties of MSC-EVs. Following in vivo MSC-EV treatment, gene set enrichment analysis (GSEA) of SBH/y-DS cortices revealed downregulation of immune system response-related gene sets. In addition, MSC-EVs downregulated gene sets related to apoptosis, wound healing and coagulation, and upregulated gene sets associated with synaptic signaling and cognition. While no specific gene was markedly altered upon treatment, the synergistic effect of all gene alternations was sufficient to increase animal survival and improve the neurological state of affected SBH/y-DS rats. Our data suggest MSC-EVs act as microenvironment modulators, through various molecular pathways. We conclude that MSC-EVs may serve as beneficial therapeutic measure for multifactorial disorders, such as CSVD.

## 1. Introduction

Cerebral small vessel disease (CSVD) includes a variety of degenerative alternations in the brain, resulting from damage to small perforating arterioles, capillaries, and venules [1]. The exact pathophysiology of CSVD is still elusive. Nevertheless, endothelial dysfunction and blood brain barrier (BBB) disruption are considered leading hallmarks in the pathological process [2]. Endothelial dysfunction and BBB disruption are involved in a cascade of pathological events including reactive oxygen species (ROS) production, nitric oxide (NO) depletion, functional and structural alterations of the blood vessels, and immune system activation [3,4]. These pathological mechanisms may cause common CSVD manifestations, including cerebral microbleeds, intracerebral hemorrhages (ICH), lacunar infarcts, white matter hyperintensities (WMH) and enlarged perivascular spaces [5]. Disease progression in the small vessels can lead to both ischemic and hemorrhagic consequences [1]. However, stroke caused by small vessel disease has rarely been methodically investigated, and recent progress in treatment and prevention of stroke mainly apply to large vessel pathology [6].

The therapeutic potential of mesenchymal stem cell (MSC)-derived small extracellular vesicles (EVs) has been the subject of research since 2013, when it was first reported that they have therapeutic effects on a central nervous system (CNS) disease [7]. EVs are nanometric cell-derived membranous structures, comprising diverse molecular cargo of proteins, lipids, RNAs and miRNAs [8,9]. Xin, et al. substantiated the claim that MSC-EVs not only cross the BBB, but can also aid in functional recovery [7]. Since then, the therapeutic capacity of EVs in various CNS pathologies has been repeatedly demonstrated [10,11]. The therapeutic property of MSC-EVs is only one aspect of the characteristics they retain from their parent cells. MSC-EVs have also been shown in several models of pathology to migrate and accumulate specifically in damaged areas within the brain [12,13], which adds further to their therapeutic value.

MSC-EVs promote functional recovery within the brain through four major mechanisms: neuroprotection, neurogenesis, neuromodulation, and angiogenesis [10]. Several studies have shown a specific chain of signals activated by the EVs through specific proteins and miRNAs incorporated in the EV cargo [14,15,16,17,18,19,20,21,22]. Knowledge of the precise underlying mechanisms of EV treatment is still lacking.

We previously described a novel model of CSVD using the salt-sensitive hypertension prone SBH/y rat model, which when salt-loaded with deoxycorticosterone-acetate (DOCA)-salt, incorporates hypertension with peripheral oxidative stress [23]. In the current study, we investigated the potential therapeutic effect of MSC-EVs in SBH/y rat model of CSVD, while uncovering mechanisms underlying CSVD, and molecular changes induced by MSC-EV treatment.

## 2. Results

### 2.1. MSC and MSC-Derived EV Characterization

Human MSCs were isolated from a human adipose biopsy, and 20 × 10^6^ cells were seeded in ADVA X3 device bioreactor. Results of flow cytometry prior to seeding of the cells demonstrated high expression of the MSC marker CD90, and very low expression of the hematopoietic marker CD45 (Appendix A).

Medium was collected at days 4, 8, 14, 18, 20 and 26, in a total volume of 2.9 L. At the peak of growth (~day 20) there were ~10^9^ cells in the bioreactor. Cells were harvested on day 26 and analyzed again for expression of CD90 and CD45. 99.7% of the cells still expressed the CD90 surface marker and were negative to CD45 (Appendix A), indicating that the cells did not differentiate as a result of the growth conditions.

MSC-EVs were isolated from the collected medium, using serial centrifugations. Nanoparticle tracking analysis (NTA) was used to characterize the EVs in terms of size and concentration (Appendix A), and transmission electron microscopy (TEM) validated EV morphology and the purity of the isolation (Appendix A). Assessment for canonical surface CD63 and CD81 marker expression, revealed that the EVs were positive for the pan-EV markers (Appendix A). Furthermore, MACSPlex analysis, applied to determine EV origin by surface protein composition [24], revealed expression of the MSC markers CD29 and CD44, in addition to the typical EVs proteins CD9, CD63, and CD81 (Appendix A). All parameters of EV production from cells cultured in a bioreactor are summarized in the table in Appendix A. These data indicate a high production potential of EVs using this method of cell growth.

### 2.2. MSC-EVs Accumulate in Cortices of SBH/y-DS

MSC-EVs have previously been shown to migrate to the site of injury in several animal models of pathology [12,13]. In order to examine migration ability of MSC-EVs in the SBH/y model of CSVD, MSC-EVs were labeled with PKH26 fluorescent dye, and subsequently intranasally administered to SBH/y-DS, one month after initiation of DOCA-salt. PKH26-labeled EVs accumulated primarily in lesion sites in the cortex (Figure 1A–E), while in the negative control PKH26-treated rats (without EVs), they were almost undetectable within the brain (Appendix AA–E). EV accumulation in the cortex coincided with the main injured brain region in SBH/y-DS. Notably, EV accumulation was proximal to reactive astrocytes, as marked by GFAP staining (Figure 1E).

### 2.3. MSC-EVs Impact Anti-Inflammatory Response In Vitro

Considering MSC-EV proximity to reactive astrocytes, and the general notion that migration of MSC-EVs is driven by inflammation [13,25], MSC-EV effects on immune system responses were assessed in vitro. Microglia BV2 cells were stimulated using lipopolysaccharide (LPS), as a model for an inflammatory environment in the brain [26]. Treatment with MSC-EVs significantly diminished expression level of the pro-inflammatory mediators *Il1β*, *Il6*, and to some extent moderated *Ccl3* expression (Figure 2A–C). MSC-EV treatment also significantly reduced the inflammatory mediated caspase, *Casp1* (Figure 2D), further supporting the anti-inflammatory activity of MSC-EVs, and reduction in pro-inflammatory cytokines in particular.

### 2.4. Uncovering Mechanisms of MSC-EV Treatment In Vivo

Human CSVD is known to involve inflammatory signaling cascade [27]. Our previous results also established elevated immune brain response in SBH/y-DS [23]. To further broaden our understanding of the immune involvement in our model, RNA from rat cortices was studied by RNA-sequencing analysis. Gene set enrichment analysis (GSEA) identified several upregulated pathways associated with immune system response in the SBH/y-DS as compared with that of SBH/y sham. These pathways include inflammatory response, immune response regulation, immune response regulating signaling pathway, innate and adaptive immune response, T cell activation, cytokine production, lymphocyte activation, and enhancement of immune system processes (Figure 3A–I).

In order to uncover molecular pathways underlying MSC-EV treatment, GSEA was used in MSC-EV treated SBH/y-DS compared to untreated DOCA-salt loaded SBH/y. The analysis revealed downregulation of immune system response-related genes, including inflammatory response, innate and adaptive immune response, T cell activation, cytokine production, lymphocyte activation, positive regulation of immune system process, interferon gamma response, TNF signaling via NFκB, and IL6-JAK-STAT3 signaling (Figure 4A–G). In addition, downregulation of genes related to apoptosis, wound healing and coagulation was observed (Figure 4K–M). GSEA identified upregulated pathways associated with synaptic signaling, regulation of trans-synaptic signaling, and cognition in MSC-EV treated SBH/y-DS (Figure 4N–P).

To validate these findings, RT-PCR was performed to identify specific mRNA changes related to the above-mentioned pathways. Levels of both *Gfap* (reactive astrocytes marker) and *Aif1* (activated microglial marker) mRNA were higher in SBH/y-DS than in SBH/y sham, and they were normalized following MSC-EV treatment, indicating anti-inflammatory activity of MSC-EVs (Figure 5A,B, respectively). The expression of cortical *Bmp7* mRNA, a member of the TGFβ superfamily, which was identified as an immune system regulator [28], was higher in the SBH/y-DS than in SBH/y sham, and decreased following MSC-EVs treatment (Figure 5C). Likewise, expression of *Ccl5*, a chemokine known to regulate T cell activation [29], was upregulated in SBH/y-DS compared to SBH/y sham, and diminished following MSC-EV treatment (Figure 5D). Additionally, the expression of transcription factor *Nfκb*, which regulates multiple aspects of innate and adaptive immune functions and serves as a pivotal mediator of inflammatory responses [30], was higher in SBH/y-DS than in SBH/y sham rats, and downregulated in the treated SBH/y-DS (Figure 5E). Moreover, the expression of AKT regulator *Phlpp1*, was significantly reduced following MSC-EV treatment in SBH/y-DS, implying diminished cell death and vascular disruption (Figure 5F) [31]. Expression of both *Serpine1* and *S100a6* mRNA was higher in DOCA-salt SBH/y than sham SBH/y, and normalized by MSC-EV treatment, suggesting anti-thrombotic and anti-neurodegenerative activities of the MSC-EVs (Figure 5G,H, respectively) [32,33]. Notably, all gene alternations, with the exception of *Phlpp1*, were not significantly different for each gene separately, although they all shared a common trend.

### 2.5. MSC-EVs Alleviate Clinical Features and Cognitive Deficits of SBH/y-DS

In order to assess the therapeutic potential of MSC-EVs in the CSVD model, DOCA and salt-loaded SBH/y were intranasally treated with MSC-EVs once a week, starting one week after DOCA administration. Clinical features of the rats were monitored over two months. The DOCA-salt loaded SBH/y groups displayed a significantly slower growth rate compared to sham-treated SBH/y from day 21 and on. Nevertheless, from day 42, MSC-EV treatment significantly improved the rat’s growth rate compared with the not-treated group (Figure 6A). This improvement is further demonstrated in the Kaplan-Meier survival curve, where only untreated SBH/y-DS had to be sacrificed due to >10% loss of body weight (Figure 6B).

Neurological deficits were assessed using a serial of double-blinded neurological examinations, as previously discussed [34]. Starting from day 36 following induction of hypertension with DOCA-salt loading, SBH/y-DS showed a steady increase in neurological scoring, which became significant on days 53 and 60 compared with SBH/y sham and MSC-EVs treated rats, respectively (Figure 6C). The neurological scoring of MSC-EVs treated rats remained comparable to SBH/y sham rats throughout the experiment.

Novel object recognition test was employed to assess recognition memory of the rats. SBH/y-DS significantly underperformed and spent less time exploring the novel object compared with both SBH/y sham and MSC-EVs treated rats (Figure 6D), indicating MSC-EV treatment improved cognitive deficits.

## 3. Discussion

The aim of this study was to determine the feasibility of MSC-EVs as a potential therapy for CSVD, while uncovering molecular pathways underlying both CSVD and MSC-EV treatment. Involvement of immune system-related pathways were exhibited in both DOCA-salt loaded SBH/y rats and MSC-EVs, yet in an opposite manner. While the cortex in SBH/y-DS demonstrated immune response upregulation, MSC-EVs normalized these pathways. In addition, MSC-EVs downregulated genes that participate in apoptosis, wound healing, and coagulation, and upregulated genes associated with synaptic signaling and cognition. These effects improved clinical features and cognitive deficits in SBH/y-DS, as demonstrated by survival rate, neurological scores, and the novel object recognition test.

In the current study, we administrated EVs intranasally to the experimental animals. Although the mechanism by which EVs cross biological barriers has not been fully characterized, it is considered to depend on the characteristics of the EVs, some of which are influenced by their cellular origin [35]. The behavior of MSC-EVs, for example, which were shown to migrate and accumulate in damaged areas within the brain, resembles that of their parent cells [13,25,36,37,38,39,40]. This feature has been demonstrated in a number of other pathological CNS model disorders [10]. In the SBH/y model of CSVD, we showed that MSC-EVs concentrate in lesion sites in the cortex, an area largely damaged in this model. Concentration of an agent at lesion sites has an advantage when searching for therapy, as this minimizes undesirable systemic effects. This feature of “targeted delivery” is extremely important for inaccessible organs, such as the brain, and particularly in disorders such as CSVD, where the damage is varied and differs between patients, and there is no single area to target.

It is widely recognized that chemotaxis and inflammation both play a role in the migration ability of MSC-EVs [13,25]. This is consistent with our model, in which EVs were mostly co-localized with GFAP staining. Moreover, we demonstrated an anti-inflammatory activity of MSC-EVs in vitro, further supporting previous studies describing immune response modulation by MSC-EVs [41,42,43,44,45,46,47,48,49,50,51,52]. Inflammation is increasingly recognized as a risk factor for CSVD through both cerebrovascular risk factors, such as hypertension and type-2 diabetes mellitus, and through immune system senescence [53]. A previous study found a significant correlation in CSVD patients with stroke, between vascular inflammatory markers and CSVD [54]. Furthermore, a longitudinal study demonstrated that baseline expression levels of systemic inflammatory factors could predict the severity of the subsequent CSVD. Several anti-inflammatory drugs and immunomodulatory agents have been proposed as a potential therapeutic intervention for CSVD [54,55]. However, clinical efficacy must still be demonstrated.

Whether inflammation causes CSVD or, is secondary to CSVD is not entirely clear [54]. Nevertheless, as CSVD is a multifactorial disorder, treatment strategies that combine drugs that incorporate multiple mechanisms of action is thought to be promising [53,55]. As the EV molecular cargo comprises thousands of proteins, lipids, RNAs and miRNAs affecting simultaneously multiple pathways, MSC-EVs are a potential candidate for CSVD treatment.

Proposed mechanisms of action of MSC-EVs include neuroprotection, neurogenesis, immunomodulation, and angiogenesis, which are mediated by the synergistic effect of their molecular cargo [45,56]. As discussed, GSEA of differential expression between MSC-EVs treated and untreated DOCA-salt loaded SBH/y rats, revealed several gene sets involved in immune system response, that were downregulated in the treated rats, further supporting immunomodulation properties of MSC-EVs. MSC-EVs also affected gene sets associated with apoptosis, wound healing, coagulation, synaptic signaling, and cognition. Affecting such varied pathways demonstrates the multifactorial approach relevant for treating CSVD. Notably, most of the gene alternations were not statistically different for each gene separately but rather as gene sets, emphasizing the synergistic effect of the EVs molecular cargo. Importantly, this synergistic effect manifested in improved clinical features and cognitive deficits of MSC-EVs treated SBH/y-DS.

We conclude that MSC-EV payload has the potential to alter the microenvironment of the target tissue in a way that could reverse pathological conditions and restore it to a healthier state. Due to an intrinsic tissue-homing capacity, MSC-EVs have the potential to be manipulated into drug delivery vehicles [9]. EVs have the potential to become multi-drug delivery platform because of their ability to secure cargo from degradation and avoid recognition and subsequent clearance by the immune system. The outcome of microenvironment modulation could lead to new therapeutic opportunities for multifactorial disorders such as CSVD.

## 4. Materials and Methods

### 4.1. Cell Culture

#### 4.1.1. Primary Culture of Human Adipose-Derived Mesenchymal Stem Cells

Human MSCs were isolated from subcutaneous adipose tissue, as previously described [57]. A fresh biopsy (one patient) was taken by a plastic surgeon during surgery performed for reasons unrelated to the biopsy for the current research. A written informed consent was obtained from the patient prior to surgery. Experimental work with the human adipose tissue was approved by the Helsinki Committee of the Israeli Ministry of Health (protocol code 0417-20, 18 June 2020).

Briefly, the adipose tissue was extensively washed with sterile phosphate-buffered saline (PBS) containing 1% penicillin-streptomycin-neomycin (PSN) (Biological Industries, Beit Haemek, Israel), and minced thoroughly, followed by an incubation with collagenase I solution for 1 h at 37 °C. After digestion, sample was diluted 1:1 in Dulbecco’s modified Eagle’s medium (DMEM) supplemented with 10% fetal bovine serum (FBS) (Biological Industries, Beit Haemek, Israel), and filtered through 100 and 40 μm filters. A filtered sample was centrifuged at 600× *g* for 7 min, and the pellet was suspended in 1 mL of basal medium, followed by incubation with 3 mL of erythrocyte lysis buffer (155 mM NH_4_Cl, 10 mM KHCO_3_, 0.1 mM EDTA) for 5 min at room temperature (RT). 10 mL of sterile PBS was added, and the sample was centrifuged at 600× *g* for 7 min. The pellet was suspended with 1 mL of basal medium, and cells were plated at a density of 30 × 10^3^ cells/cm^2^ in MSC growth medium, containing DMEM, 10% FBS, 2 mM L-glutamine, 1% PSN, and 1% non-essential amino acids (NEAA) (Biological Industries, Beit Haemek, Israel). FBS was depleted of EVs by ultracentrifugation for 18 h at 100,000× *g* using a 45 Ti rotor (Beckman, Pasadena, CA, USA). Adherent cells were cultured to 90% confluence and then reseeded. When cell concentration reached 20 × 10^6^ cells, they were seeded in ADVA X3 bioreactor (Adva Biotechnology, Bar Lev Industrial Park, Israel) to achieve scale up EV production. Cells were grown in serum-free medium (ATCC), and medium was collected at days 4, 8, 14, 18, 20 and 26. Collected medium was stored at −80 °C. Cells were maintained at 37 °C in a humidified 5% CO_2_ incubator.

#### 4.1.2. Lipopolysaccharide (LPS)-Induced Microglia Activation

Mouse microglia cells, BV2, were grown in suspension in 75 cm^2^ culture flasks (Corning, Corning, NY, USA) in DMEM, supplemented with 10% FBS, 2 mM L-glutamine and 1% PSN antibiotics (Biological Industries, Beit Haemek, Israel). Cells were grown to 70–80% confluence before experiments were conducted, and maintained at 37 °C in a humidified atmosphere containing 5% CO_2_.

For lipopolysaccharide (LPS)-induced microglia activation, 300,000 BV2 cells were seeded in 6-well plate. Microglia activation was induced by exposing cells to 1 µg/mL LPS (Sigma-Aldrich, Rehovot, Israel), a non-infectious component of Gram-negative bacterial cells wall, for 20 h. In parallel to LPS stimulation, 10^10^ MSC-EVs were added to the culture media. Anti-inflammatory activity was then evaluated using RT-PCR for pro-inflammatory cytokines.

### 4.2. Extracellular Vesicle (EV) Isolation

EV isolation was accomplished using a standard differential centrifugation protocol as described previously [58]. Medium was centrifuged for 10 min at 300× *g*. The resulting supernatant was centrifuged for 10 min at 2000× *g*. Once again, the supernatant was recovered and centrifuged for 30 min at 10,000× *g*. The supernatant was recovered and filtered through a 0.22 µm filter and centrifuged for 2 h at 100,000× *g*. Each centrifugation was conducted at 4 °C. The supernatant was discarded, and the pellet was suspended in sterile PBS and filtered using an Amicon centrifugal filter (30 kda, Ultracel-30 regenerated cellulose membrane, 0.5 mL Amicon filter, Millipore Corp., Bedford, MA, USA) to remove soluble protein contamination. The pellet, containing purified EVs, was suspended in sterile PBS, aliquoted and frozen at −80 °C for further experiments and characterization.

### 4.3. Flow Cytometry

#### 4.3.1. MSC Characterization

MSCs were trypsinized and centrifuged at 1000× *g* for 5 min, followed by an incubation with CD90-FITC (cat#130-114-859, Miltenyi biotec, Bergisch Gladbach, Germany), CD45-APC (cat# 130-110-633 Miltenyi biotec) or IgG1 Isotype control (cat#130-113-434, Miltenyi biotec) fluorescent antibodies for 30 min on ice (10^6^ cells/sample). 700 µL sterile PBS was added, and the mixture was centrifuged at 1000× *g* for 5 min. The pellet was suspended in 400 µL sterile PBS, and samples were analyzed by flow cytometry using Gallios flow analyzer FACS (Beckman Coulter, Brea, CA, USA). Data were analyzed using the Kaluza Analysis Software (Beckman Coulter).

#### 4.3.2. EV Characterization

EVs were coated onto 4-μm-diameter aldehyde/sulfate latex beads. Moreover, 50 µL EVs were incubated with 12.5 µL 4-μm-diameter aldehyde/sulfate latex beads (cat# A37304, Invitrogen, Carlsbad, CA, USA) for 15 min at RT. 700 µL sterile PBS was added, and the mixture was transferred to 4 °C, with gentle overnight shaking. After centrifugation, the pellet was blocked by incubation with 200 µL 100 mM glycine for 30 min at RT. EV-coated beads were washed in PBS and suspended in 100 µL sterile PBS. The beads were then incubated with CD63-APC (cat#130-127-492, Miltenyi biotec), CD81-APC (cat# 130-119-787 Miltenyi biotec) or IgG1 Isotype control (cat#130-113-434, Miltenyi biotec) fluorescent antibodies for 15 min on ice in the dark. Beads were analyzed by flow cytometry using Gallios flow analyzer FACS (Beckman Coulter). Data were analyzed using the Kaluza Analysis Software v.2.1 (Beckman Coulter).

### 4.4. Nanoparticle Tracking Analysis (NTA)

EVs were subjected to NTA with Nanosight NS300 instrument (NanoSight Ltd., Malvern, UK) to determine particle size and concentration. Size distribution and particle concentration were measured by capturing the particles undergoing Brownian movement, then analyzed with NTA 3.0 software (Malvern Instruments Ltd, Malvern, UK). Each sample was measured five times and video images were captured simultaneously. In each measurement, size distribution and concentration were averaged.

### 4.5. Transmission Electron Microscopy (TEM)

For negative staining electron microscopy, EV pellets were suspended in sterile PBS after purification. Samples were adsorbed on Formvar/carbon-coated grids and stained with 2% aqueous uranyl acetate. They were then examined using a JEM 1400plus transmission electron microscope (Jeol, Tokyo, Japan). Images were captured using SIS Megaview III and iTEM, the imaging platform for transmission electron microscopy (Olympus, Tokyo, Japan).

### 4.6. Multiplex Surface Marker Analysis

MACSPlex analysis was performed using the MACSPlex Exosome, human, Kit (#130-108-813, Miltenyi Biotec) according to kit specifications. The overnight capture antibody incubation protocol was applied, with a combination of CD9-, CD63- and CD81-APC antibodies. Flow cytometric analysis was carried out on a Gallios flow analyzer FACS (Beckman Coulter). Data were analyzed using the Kaluza Analysis Software (Beckman Coulter). For further analysis, background values of the blank sample (PBS) of each run were subtracted from the sample values. Each surface marker median intensity was normalized to CD9-, CD63- and CD81-APC antibodies signal (positive EVs).

### 4.7. Animals

Male Sabra hypertension-prone rats (SBH/y) were obtained from the Israeli Rat Genome Center at the Barzilai University Medical Center (Ashkelon, Israel) [59]. Rats were maintained in 12-h-light/12-h-dark conditions in acclimatized rooms and had free access to food and water. All experimental protocols were authorized by the Tel Aviv University Committee of Animal Use for Research and Education. Every effort was made to reduce the number of rats used and minimize their suffering.

#### Experimental Design and Induction of Hypertension

Animals were weaned at age six weeks and randomly assigned to experimental or control groups (SBH/y-DS that were treated with MSC-EVs (*n =* 14), and SBH/y-DS that were untreated with MSC-EVs (*n =* 13), respectively). We salt-loaded the animals to elicit hypertension in SBH/y by implanting subcutaneously a 60-day release 75 mg deoxycorticosterone-acetate (DOCA) pellet (Innovative Research of America) and provided with 1% NaCl and 0.1% potassium chloride in tap water. Isoflurane was used for both induction (5%) and maintenance (1.5–2%) of anesthesia. We have previously shown that SBH/y-DS invariably developed hypertension after 1 month that persists as long as the DOCA pellet continues to secrete deoxycorticosterone acetate and NaCl is provided in drinking water [23]. Sham control animals (“SBH/y sham” group (*n =* 11)) were subjected to anesthesia, skin incision and suturing without insertion of the DOCA pellet, and rats were provided tap water with no added salt or potassium.

Animals were monitored daily for two months. Animals in the experimental but not in the sham group were administered 35 µL of MSC-EVs intranasally once a week, starting one week after DOCA implantation (total of 1.4 × 10^11^ EVs per rat).

If signs of stroke developed or a loss of >10% body weight/week occurred and it appeared that death was imminent, the animal was sacrificed. Sacrificed animals were incorporated in the survival curves. Their tissues were harvested and analyzed.

### 4.8. EV Labeling

EVs were labeled with PKH26 (Sigma-Aldrich). 100 μL of EVs (1.2 × 10^9^ particles/µL) or 100 μL PBS (negative control- free PKH26) were added to 1 mL Diluent C. PKH26 (6 μL) was then added for 5 min of incubation in RT. Quenching was done by adding 2 mL 10% bovine serum albumin (BSA) in PBS. EVs or negative control were suspended in 70 mL PBS and centrifuged for 120 min at 100,000× *g* at 4 °C. The pellet was suspended in a total of 60 μL of PBS.

6 SBH/y-DS rats were treated with PKH26-labeled EVs (*n =* 3) or negative control (free PKH26, *n =* 3) 30 days after initiation of salt-loading and induction of hypertension. EVs or free PKH26 were delivered intranasally (20 µL) to both sides of the nasal cavity (10 µL to each side). 24 h post-administration, rats were anesthetized with Ketamine/Xylazine and transcardially perfused with cold PBS followed by 4% PFA-PBS. The brain was surgically removed, immersed in 4% PFA for 24 h at 4 °C, followed by cryoprotection in 30% sucrose solution for an additional 48 h and stored at 4 °C in PBS supplemented with 0.01% sodium azide until section preparation. For immunostaining, brains were embedded in OCT, frozen on dry ice, cut in coronal sections (20 μm) using a freezing sliding microtome (Leica CM1850) and stored at –20 °C until use.

### 4.9. Histological Staining

Three slices per brain stained with hematoxylin-eosin (Sigma-Aldrich) using standard protocols were used to identify cerebral edema, ICH, and lacunar infarcts.

### 4.10. Immunohistochemistry

Slides were thawed, washed twice with PBS, blocked and permeabilized with PBS 1% BSA, 5% goat serum (Biological Industries, Galilee, Israel) and 0.05% Triton-X (Sigma-Aldrich) for one hour, and incubated with GFAP (1/1000, ab7260, Abcam, Cambridge, UK) primary antibody overnight at 4 °C. The slides were subsequently washed 3 times with PBS and incubated with a fluorescent-labelled secondary antibody (1/700, Alexa-Flour) for 1 h at room temperature. DNA was stained for 10 min with DAPI (1:1000, Sigma-Aldrich), and the slides were mounted using Flouromount-G.

### 4.11. Microscopy and Image Analysis

Slides stained for histology or immunohistochemistry were either imaged using the Olympus BX52 microscope (Olympus America Inc., Allentown, PA, USA) or using the AxioImager Apotome microscope (Zeiss, Jena, Germany). Image analysis was performed using ImageJ software. Analyses parameters in ImageJ were maintained constant for every staining in all the slides, in order to avoid bias and exclude background interference.

### 4.12. Neurological Deficit Scoring

The method for neurological assessment was adapted and modified from Hunter, et al. [34], and comprised serial examinations, including assessment of paw placement and flexion, ability to grip and stabilize on a horizontal bar, visual forepaw reaching, contralateral rotation, Pinna and corneal reflexes, general condition, motility, limpness, and circling. Scoring was performed twice a week from day 30 of the study until termination. In the animals that had to be sacrificed along the study, the clinical scores were fixed as the last score prior to sacrifice.

### 4.13. Novel Object Recognition (NOR) Test

For acclimation prior to testing the animals, they were transferred to a dedicated behavioral testing room 30 min prior to the beginning of each study. Chambers were cleaned with Virusolve+ between animal testing. Tests were monitored and analyzed using an automated tracking system (Ethovision v11.5, Noldus, Wageningen, Netherlands).

The test was conducted in an open plastic chamber (50 × 50 × 50 cm) with white walls and a black floor. One week prior to testing, rats were trained by allowing them to explore the box for 10 min. During the training session on experimental day 49, two identical objects were placed in the chamber. Twenty-four hours later, one of the familiar objects was substituted with a novel object, and rats were allowed to explore the chamber again for 5 min and we calculated and recorded the time spent sniffing the novel and familiar objects.

### 4.14. RNA Extraction

Rats were anesthetized with CO_2_ and decapitated. The brains were immediately removed, and cortices were snap-frozen in liquid nitrogen and stored at –80 °C. RNA was extracted using PureLink™ RNA Mini Kit (Thermo Fisher scientific, Waltham, PA, USA) according to manufacturer’s instructions.

### 4.15. RNA Libraries Preparation and Sequencing

RNA-seq libraries were prepared from 500 ng of RNA, using the NEBNext Ultra II Directional RNA Library Prep Kit for Illumina (New England Biolabs Inc., Ipswich, MA, USA, E7760S) with the NEBNext Poly(A) mRNA Magnetic Isolation Module (New England Biolabs Inc., Ipswich, MA, USA, E7490S), following manufacturer’s instructions. All libraries were amplified using 11 PCR cycles. Resulting libraries were quantified and analyzed by Qubit (4, Life Technologies, Carlsbad, CA, USA, extended range XR kit #Q33223) and TapeStation (4200, CAT #G2991BA). Libraries were sequenced on the NextSeq 500 platform (Illumina, San Diego, CA, USA), following the manufacturer’s protocol, using a NextSeq 500/550 High Output Kit version 2.5 (75 cycles) (Illumina). Sequencing was performed at the Genomics Research Unit at the Life Sciences Inter-Departmental Research Facility Unit, Tel Aviv University.

### 4.16. RNA-Seq Bioinformatics Analysis

12 FastQ files were obtained for the three groups and uploaded to Partek^®^ Flow^®^ data analysis software for processing (build version 9.0.20.0804). Reads were trimmed for low-quality bases from the 30′ end of reads (Phred < 20). Average of read >45 million per sample, and read quality was high. Alignment was performed using STAR–2.7.3a to rat genome rn7 [60]. Aligned reads were filtered for low-quality mapping (Phred < 20). Quantification of reads was performed using the Partek E/M algorithm, detecting 34,726 genes [61]. DESeq2 median ratio normalization was performed following DESeq2 analysis of differential expressed genes (with cutoff *p* < 0.05 and fold-change (FC) difference = 2) [62]. Analysis and graphs were constructed by the Partek^®^ Flow^®^ and Partek GS (Partek Genomics Suite v 7.19.1125, St. Louis, MO, USA) software.

### 4.17. Gene Set Enrichment Analysis (GSEA)

Whole differential expression data were subjected to gene set enrichment analysis using GSEA [63], which uses all differential expression data (cutoff independent) to determine whether a priori defined sets of genes show statistically significant, concordant differences between two biological states. The following gene set collections: Hallmark and gene ontology (biological process) from the molecular signatures database (MSigDB) were used.

### 4.18. Real-Time PCR

RNA was reverse transcribed to complementary DNA (cDNA) using iScript cDNA Synthesis Kit (BIO-RAD). Semi-quantitative PCR was performed on the Step-One real time PCR (RT-PCR) system using Platinum SYBR Green (Invitrogen) and the custom designed primers. Threshold cycle values were determined in triplicates and presented as average compared with Actin. Fold changes were calculated using the 2∆CT method.

### 4.19. Statistical Analyses

All statistical analyses were performed using GraphPad Prism 6. Data are expressed as mean ± SEM. Neurological scores and growth curve were analyzed using a two-way ANOVA followed by Tukey’s post hoc analyses. NOR was analyzed using an unpaired student’s *t*-test. Statistical analysis of RT-PCR was performed using one-way ANOVA followed by Tukey’s post hoc analyses. For all tests, the statistical significance threshold was set to *p* < 0.05.

The animal study protocol was approved by the Committee of Animal Use for Research and Education of Tel Aviv University (protocol code 01-20-026, 11 May 2020).

## Figures and Tables

**Figure 1 ijms-23-11211-f001:**
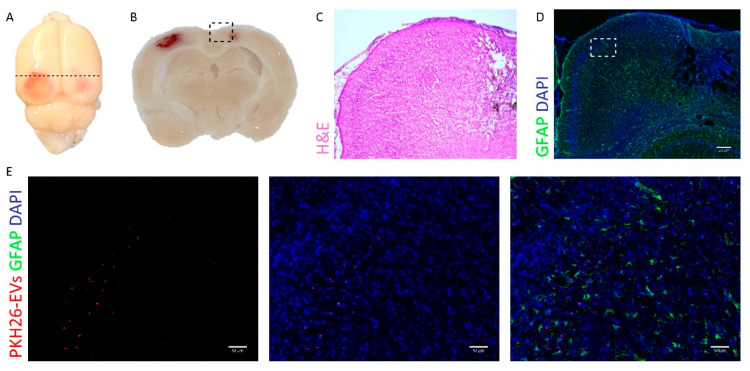
PKH26-labeled extracellular vesicles (EVs) accumulate in lesion sites in brains of the SBH/y model of cerebral small vessel disease (CSVD). Representative images of brain derived from SBH/y-DS rats following a single administration of PKH26-labeled EVs (**A**). Dashed lines represent the coronal section presented in (**B**). 5× magnification of the injured area is presented using H&E (**C**) and GFAP immunostaining (**D**). Scale bar = 200 µm. A magnification of the inset in (**D**) is presented in (**E**). PKH26 labeled EVs are shown in red. Scale bar = 50 µm.

**Figure 2 ijms-23-11211-f002:**
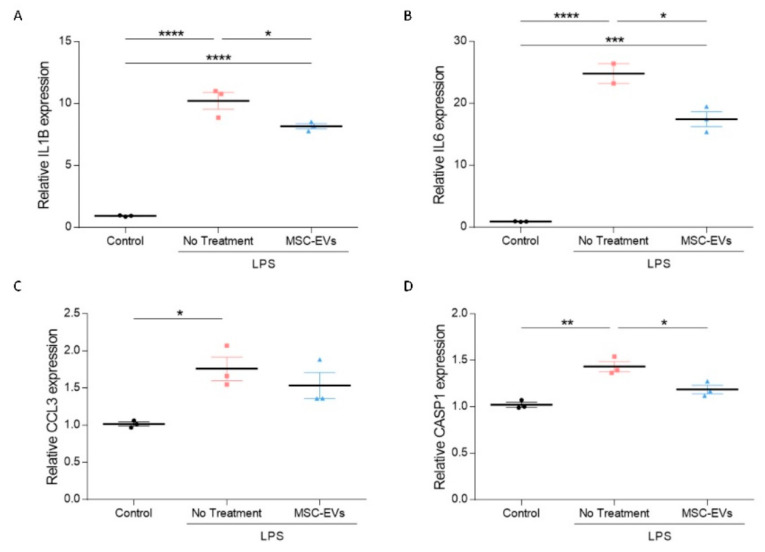
Mesenchymal stem cell-derived extracellular vesicles (MSC-EVs) have an anti-inflammatory effect on activated BV2 microglia cells. mRNA expression analysis of *Il1β* (**A**), *Il6* (**B**), *Ccl3* (**C**), and *Casp1* (**D**) in control or lipopolysaccharide (LPS)-activated BV2 cells with and without MSC-EV treatment (primers are listed in Table 1). Data are presented as mean ± SEM, * *p* < 0.05, ** *p* < 0.01, *** *p* < 0.001, **** *p* < 0.0001, one-way ANOVA.

**Figure 3 ijms-23-11211-f003:**
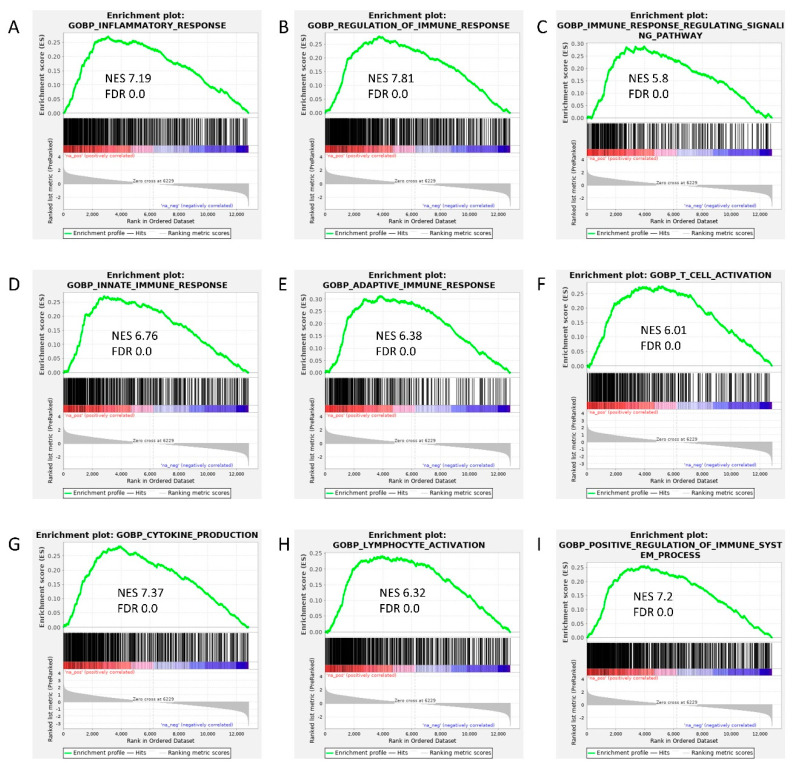
Gene Set Enrichment Analysis (GSEA) in the SBH/y model of cerebral small vessel disease (CSVD). GSEA of differential expression between SBH/y- DS (*n =* 4) and SBH/y sham (*n =* 4) demonstrates that the following gene sets were upregulated in our model: inflammatory response (**A**), regulation of immune response (**B**), immune response regulating signaling pathway (**C**), innate (**D**) and adaptive (**E**) immune response, T cell activation (**F**), cytokine production (**G**), lymphocyte activation (**H**), and positive regulation of immune system process (**I**). Green curves show the accumulating enrichment score. Normalized enrichment score (NES) and false discovery ratio (FDR) are shown for each gene set analyzed.

**Figure 4 ijms-23-11211-f004:**
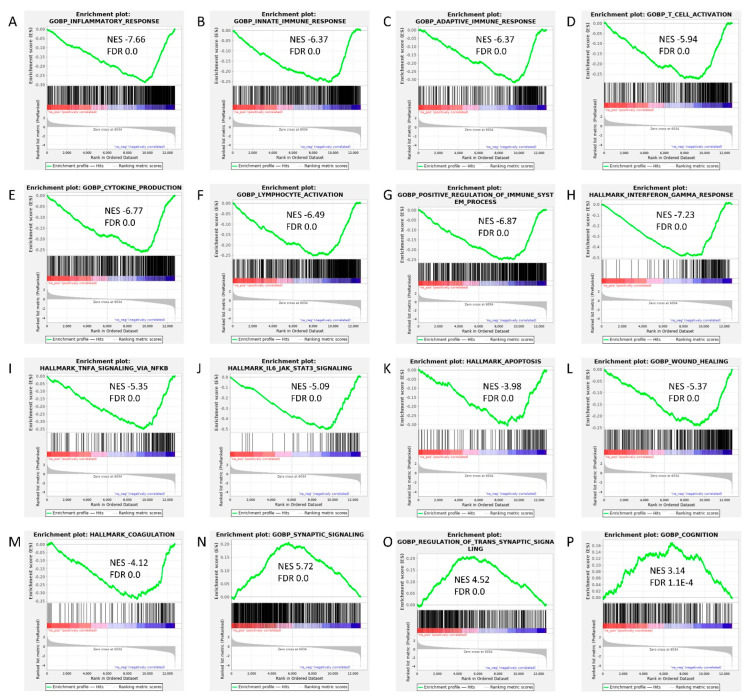
Gene Set Enrichment Analysis (GSEA) in mesenchymal stem cell-derived extracellular vesicle (MSC-EV)-treated SBH/y-DS rats. GSEA of differential expression between MSC-EV treated (*n =* 4) and untreated (*n =* 4) SBH/y-DS rats demonstrates that the following gene sets were downregulated in the treated rats: inflammatory response (**A**), innate (**B**) and adaptive (**C**) immune response, T cell activation (**D**), cytokine production (**E**), lymphocyte activation (**F**), positive regulation of immune system process (**G**), interferon gamma response (**H**), TNF signaling via NFκB (**I**), IL6-JAK-STAT3 signaling (**J**), apoptosis (**K**), wound healing (**L**), and coagulation (**M**). Furthermore, the following gene sets were upregulated in the treated rats: synaptic signaling (**N**), regulation of trans-synaptic signaling (**O**), and cognition (**P**). Green curves show the accumulating enrichment. Normalized enrichment score (NES) and false discovery ratio (FDR) are shown for each gene set analyzed.

**Figure 5 ijms-23-11211-f005:**
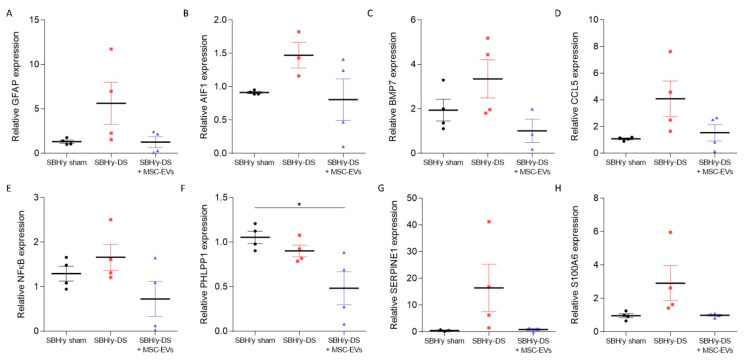
mRNA alternations between SBH/y sham, SBH/y-DS, and mesenchymal stem cell-derived extracellular vesicle- (MSC-EV) treated SBH/y-DS rats. mRNA expression analysis of *Gfap* (**A**), *Aif1* (**B**), *Bmp7* (**C**), *Ccl5* (**D**), *Nfκb* (**E**), *Phlpp1* (**F**), *Serpine1* (**G**), and *S100a6* (**H**) in rats’ cortices (*n =* 4) (primers are listed in Table 1). Data are presented as mean ± SEM, * *p* < 0.05, one-way ANOVA.

**Figure 6 ijms-23-11211-f006:**
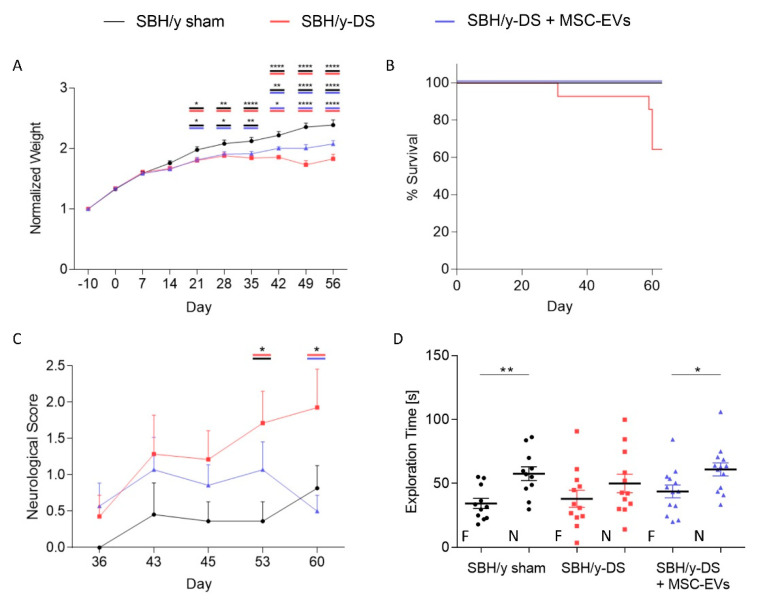
Mesenchymal stem cell-derived extracellular vesicle- (MSC-EV) treatment improved clinical features and cognitive deficits in SBH/y-DS rats. Growth rate of the animals normalized to baseline body weight (**A**). Kaplan-Meier survival curve (**B**). Neurological score assessment starting from day 36 was performed using serial neurological examinations, including assessment of walking, forelimb and hindlimb flexion, balance beam, placing test, corneal and startle reflex, myoclonus, myodystonia and shaking (**C**). The novel object recognition (NOR) task was performed to evaluate recognition memory (**D**). The time spent exploring the familiar (F) and novel (N) objects is presented. Data are presented as mean ± SEM, (**A**,**C**) * *p* < 0.05, ** *p* < 0.01, **** *p* < 0.0001, two-way ANOVA; (**D**) * *p* < 0.05, ** *p* < 0.01, two-tailed *t*-test.

**Table 1 ijms-23-11211-t001:** Primers for RT-PCR.

Specie	Gene	Sequence
Mouse	CASP1	Fw: TGTGACTTGGAGGACATTTTCAG Rev: GGTCACCCTATCAGCAGTGG
Mouse	CCL3	Fw: GTACCATGACACTCTGCAACC Rev: GTCAGGAAAATGACACCTGGC
Mouse	GAPDH	Fw: CATGGCCTTCCGTGTTCCTA Rev: CTGGTCCTCAGTGTAGCCCAA
Mouse	IL1β	Fw: GGAGAACCAAGCAACGACAAAATA Rev: TGGGAACTCTGCAGACTCAAAC
Mouse	IL6	Fw: ATGGATGCTACCAAACTGGAT Rev: TGAAGGACTCTGGCTTTGTCT
Rat	AIF1	Fw: AGCCCAACAGGAAGAGAGGT Rev: TGCTGTACTTGGGATCATCG
Rat	BMP7	Fw: ATAATTCGGCGCCCATGTTC Rev: AAACCGGAACTCTCGATGGT
Rat	CCL5	Fw: TGCCCACGTGAAGGAGTATT Rev: ACTTCTTCTCTGGGTTGGCA
Rat	GAPDH	Fw: GGTGCTGAGTATGTCGTGGA Rev: CGGAGATGATGACCCTTTTG
Rat	GFAP	Fw: GGTGGAGAGGGACAATCTCA Rev: CAGCCTCAGGTTGGTTTCAT
Rat	NFκB	Fw: AAAATATTCACCTGCACGCCC Rev: ATCCGTGCTTCCAGTGTTTC
Rat	PHLPP1	Fw: ACACATGGCTTATAACCGGC Rev: TGGTTGTTGGGATGGCTTTC
Rat	S100A6	Fw: TCTTCCACAAGTACTCTGGCA Rev: TGTTACGGTCCAGATCATCCA
Rat	SERPINE1	Fw: AGGCCTCCAAAGACCGAAAT Rev: TGAAGAAGTGGGGCATGAAG

## Data Availability

The datasets generated and analyzed during the current study are available from the corresponding author on reasonable request.

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
