# Peer review of "Mesenchymal Stem Cell-Derived Extracellular Vesicles as Proposed Therapy in a Rat Model of Cerebral Small Vessel Disease"

_ijms, 2022, doi:10.3390/ijms231911211_

Round 1

Reviewer 1 Report

The authors from "Mesenchymal Stem Cell-derived Extracellular Vesicles as Proposed Therapy in a Rat Model of Cerebral Small Vessel Disease" nicely described their results in this article. In my opinion, they covered all the necessary experiments to test their hypothesis and also brought a thoughtful discussion about mesenchymal stem cell-derived extracellular vesicle therapy. I have no alterations to suggest.

Author Response

The authors would like to thank the reviewer for the kind words.

Author Response

Major:

1.     DOCA: since the model includes a potent glucocorticoid, which is known to be highly immune suppressive, and all of the gene differences were related to the immune system, how can the authors exclude some strange artefacts? Is there a way to control or confirm the blood levels of the DOCA?

Following the reviewer's important comments, we thoroughly researched the literature and were unable to find evidence that DOCA, which is primarily considered a mineralocorticoid (and not a glucocorticoid1), has a strong effect that suppresses the immune system. Moreover,  a previous study demonstrated kidney T-cell infiltration and urinary IFN-γ excretion following DOCA- and salt-induced hypertension in rats2.

Since DOCA blood level is not a trivial measurement, we focused on the DOCA biological effects. The experiment included both SBH/y sham and SBH/y-DS groups which enable to examine the effect of DOCA-salt per se. As demonstrated in our results, DOCA-salt caused for several upregulated pathways associated with immune system response in the SBH/y-DS as compared with that of SBH/y sham.

1.         Vinson, G. P. The mislabelling of deoxycorticosterone: making sense of corticosteroid structure and function. J. Endocrinol. 211, 3–16 (2011).

2.         Moes, A. D. et al. Mycophenolate Mofetil Attenuates DOCA-Salt Hypertension: Effects on Vascular Tone. Front. Physiol. 9, (2018).

2.     Species barrier/immune reaction: Administering human cell-derived products should generate some immune responses in rats (even with a glucocorticoid, right?). With EVs containing human proteins, hormones, mRNAs, microRNAs etc. some of those may have similar functions in rat, but not all. Human antigens should induce the rat immune system in many ways, especially with administering over the nose epithelium. This said, the negative control for EVs may also be some human cell debris from the human cell culture (i.e. no functional EVs). Not sure if the depletion approach to remove fetal bovine serum (FBS) with ultracentrifugation really works in terms of removing all antigens from a third species, i.e. “bovine” – or if this adds to the potential immunogenicity problem of human antigens in a rat.

Thank you for this comment. 

Extracellular vesicles have the ability to evade recognition and subsequent clearance by the immune system3. Saleh et al. (2019) investigated whether exposure of cells to EVs could induce inflammatory reactions in recipient cells4. When HepG2 cells were incubated with EVs for 24 h there was no significant increase in the secretion of inflammatory mediators including IL-6, IL-8, IFN-γ, MCP-1 at any dose of EVs measured. Similarly, a single I.V. administration of 5 × 1010 Expi293F-derived EVs (human source) in BALB/C mice in vivo did not raise any concerns with respect to the general toxicity and cytokine levels. Necropsy and histopathology examination revealed no treatment related gross lesions or histopathology changes in tissues including brain, heart, lungs, liver, kidney, pancreas, spleen, skeletal muscle (hind leg), thymus, mesenteric lymph node, duodenum, caecum, or tail vein4. Zhu et al. (2017) who investigated repeated treatment with EVs reported that repeated systemic administrations of EVs derived from HEK293 cells (human source) over three weeks did not produce any signs of immunogenic related toxic effects in the mice5.

Furthermore, mesenchymal stem cells (MSC) are considered immuneprivileged, with low immunogenicity. They express very low levels of MHC class I, no MHC class II and do not induce activation of allogeneic lymphocytes6,7

Taking together, it is not surprising that MSC-EVs would also be immunoprivileged and would not induce significant immune reaction. In fact, this was demonstrated in a study, where human MSC exosomes were infused into immunocompetent mouse model of acute myocardial ischemia, showing therapeutic efficacy without obvious adverse effects8,9

Regarding the EV-depleted FBS: Comparison of common isolation protocols suggested that an 18-hour centrifugation period eliminates approximately 95% of RNA-containing FBS EVs, and strongly reduce the functionality of FBS vesicles10. Although not exactly pure, and since no “depletion protocol” was proven to lead to total FBS-EV elimination, we used the most common method of isolation.

3.         Samanta, S. et al. Exosomes: new molecular targets of diseases. Acta Pharmacol. Sin. 39, 501–513 (2018).

4.         Saleh, A. F. et al. Extracellular vesicles induce minimal hepatotoxicity and immunogenicity. Nanoscale 11, 6990–7001 (2019).

5.         Zhu, X. et al. Comprehensive toxicity and immunogenicity studies reveal minimal effects in mice following sustained dosing of extracellular vesicles derived from HEK293T cells. J. Extracell. Vesicles 6, 1324730 (2017).

6.         Schu, S. et al. Immunogenicity of allogeneic mesenchymal stem cells. J. Cell. Mol. Med. 16, 2094–2103 (2012).

7. Gothelf, Y. et al. Safety of repeated transplantations of neurotrophic factors-secreting human mesenchymal stromal stem cells. Clin Transl. Med. 10, 3-21 (2014).
8.         Lai, R. C. et al. Exosome secreted by MSC reduces myocardial ischemia/reperfusion injury. Stem Cell Res. 4, 214–222 (2010).

9.         Lai, R. C. et al. Derivation and characterization of human fetal MSCs: An alternative cell source for large-scale production of cardioprotective microparticles. J. Mol. Cell. Cardiol. 48, 1215–1224 (2010).

10.       Shelke, G. V., Lässer, C., Gho, Y. S. & Lötvall, J. Importance of exosome depletion protocols to eliminate functional and RNA-containing extracellular vesicles from fetal bovine serum. J. Extracell. Vesicles 3, 10.3402/jev.v3.24783 (2014).

Minor:

We agree with all of the remarks. The MS was corrected accordingly.

Round 2

Reviewer 2 Report

The authors clearly addressed the reviewer‘s concerns with many relevant references. They also included the (minor) proposals for correction. Although the reviewer is still not fully convinced of the immune evasive function of allogenic MSC-EVs here, and the additional source of proteins from another species may also interfere with the immune system, the authors provide an interesting approach for future therapies in human. With all the different caveats of interfering with the immune system and many cross-species uncertainties (which could have included in the discussion more carefully), this appears to be a publishable contribution to the field – probably to be improved or challenged in the future.